# Defluorosilylation of fluoroarenes and fluoroalkanes

Benqiang Cui[1], Shichong Jia[1], Etsuko Tokunaga[1] & Norio Shibata [1,2]

Direct activation of carbon–fluorine bonds (C–F) to introduce the silyl or boryl groups and generate valuable carbon–silicon (C–Si) or carbon–boron (C–B) bonds is important in the development of synthetically useful reactions, owing to the unique opportunities for further derivatization to achieve more complex molecules. Despite considerable progress of C–F bond activation to construct carbon–carbon (C–C) and carbon–heteroatom (C–X) bond formation, the defluorosilylation via C–F cleavage has been rarely demonstrated. Here, we report an *ipso*-silylation of aryl fluorides via cleavage of unactivated C–F bonds by a Ni catalyst under mild conditions and without the addition of any external ligand. Alkyl fluorides are also directly converted into the corresponding alkyl silanes under similar conditions, even in the absence of the Ni catalyst. Applications of this protocol in late-stage defluorosilylation of potentially bioactive pharmaceuticals and in further derivatizations are also carried out.

[1] Department of Nanopharmaceutical Sciences and Department of Life Science and Applied Chemistry, Nagoya Institute of Technology, Gokiso, Showa-ku, Nagoya 466-8555, Japan. [2] Institute of Advanced Fluorine-Containing Materials, Zhejiang Normal University, 688 Yingbin Avenue, 321004 Jinhua, China. Correspondence and requests for materials should be addressed to N.S. (email: nozshiba@nitech.ac.jp)

Organic compounds that contain a carbon–fluorine (C–F) bond are attracting increasing attention due to their remarkable applications in pharmaceutical, agrochemical, and organic materials science, which arise predominantly from the intrinsic properties of fluorine (F)[1]. Numerous efforts have hence focused on the development of effective fluorination methods, significantly expanding the diversity of available F-containing compounds[2]. With the steady progress of methods for the preparation of organofluorine compounds, the activation of inert C–F bonds has recently drawn increasing attention, especially with respect to the effective derivatization of organic fluorine compounds[3–5]. Compared to the relatively activated C–X bonds of halogenated arenes and their equivalents (Ar–X; X = Cl, Br, I, OTf, or OMs), which easily undergo oxidative addition in metal-catalyzed coupling reactions, the cleavage of C–F bonds in fluoroarenes (Ar–F) is in general significantly more challenging due to their high bond dissociation energy; they are arguably the strongest bonds that carbon can form (Fig. 1a)[6]. Furthermore, the highly inert character of fluoroarenes renders them a more favorable platform for functionalization than any other reactive arenes Ar–X[2]. Moreover, a SciFinder search revealed that fluoroarenes are the largest group of commercially available halogenated arenes; the number of registered compounds is as follows; Ar–F (6,336,383), Ar–Cl (6,186,473), Ar–Br (3,407,354), and Ar–I (433,556). Thus, not surprisingly, significant research efforts have been directed toward C–F-cleavage protocols in order to develop synthetic strategies for complex aromatic compounds from ubiquitous fluoroarenes.

Efforts on transition-metal-catalyzed cross-coupling reactions of fluoroarenes have largely been confined to C–C bond-formation reactions (Fig. 1b, path a)[7–10], while methods that employ fluoroarenes have been extended toward the formation of carbon–heteroatom bonds (C–X; X = NR$_2$, OR, SR, BR$_2$) (Fig. 1b, path b)[11–13]. One of the recent significant achievements in this area is the defluoroborylation independently reported by the groups of Martin and Hosoya; however, this method still is limited since it inevitably requires the use of electron-rich and expensive ligands such as phosphines or N-heterocyclic carbenes (NHCs), as well as relatively high temperatures (Fig. 1b, path b)[14,15]. Thus, the development of functionalization reactions for the inert C–F bonds of fluoroarenes under very mild conditions remains a challenge.

Organosilicon compounds have been extensively used in catalysis as well as a variety of other reactions as substrates and reagents. The straightforward functionalization of aryl and alkyl silanes by various organic transformations, such as the Hiyama coupling and Fleming–Tamao oxidation, is a fundamental research area. Organosilicon reagents, exemplified by trialkylsilyl cyanides (R$_3$SiCN), azides (R$_3$SiN$_3$), ally silanes (R$_3$SiCH$_2$CH=CH$_2$), and Ruppert–Prakash reagent (Me$_3$SiCF$_3$), are powerful tools for the direct-transfer functionalization of target molecules. The application of organosilicon compounds in organic electronics, photonics, and biologically active molecules for drug discovery has also been explored owing to their useful physicochemical properties. While a wide range of methods for the synthesis of organosilicon compounds have been developed, methods for the transformation of fluoroarenes into aryl silanes do not exist. Prompted by the versatility and pivotal role of organosilicon compounds[16], as well as by our ongoing interest in the activation of inert C–F bonds[17], we report herein an available example of the ipso-silylation of aryl and alkyl fluorides via cleavage of unactivated C–F bonds under catalytic conditions mediated by Ni or in the absence of metal catalyst (Fig. 1c). By means of this process, we have prepared a broad variety of aryl silanes and alkyl silanes. Since silicon and fluorine form one of the strongest bonds in organic chemistry (Si–F: 565 kcal/mol), the cleavage of C–F bonds using silylated reagents via formation of Si–F bonds is a very common strategy. Hence, the present method is an extremely rare example of a defluorosilylation reaction using silylated reagents.

## Results

**Optimization study.** Initially, we chose 4-fluorobiphenyl (**1a**) as the substrate and a Ni catalyst to examine the C–F bond-cleavage reaction with the easily accessible silyl boronate **2a** (Table 1; Supplementary Tables 1–7)[18]. Contrary to our initial expectation of the formation of an Aryl-Bpin compound, i.e., p-Ph-Ph-Bpin, we discovered that **1a** was converted into silyl arene **3a** (50% GC yield) using Ni(COD)$_2$ (10 mol%; COD = cyclooctadiene), IPr·HCl (20 mol%), and NaOtBu (3 equiv) (toluene; temp = 110 °C; t = 24 h; Supplementary Table 1). The corresponding nucleophilic aromatic substitution (S$_N$Ar) product from NaOtBu was not observed either. Encouraged by this highly unexpected result, we decided to focus on this defluorosilylation of fluoroarenes. After screening the reaction conditions (Supplementary Tables 1 and 2), a moderate yield of **3a** was obtained in the absence of an exogenous ligand (entry 1, Table 1). Even though NHC and phosphine ligands can be effective for the activation

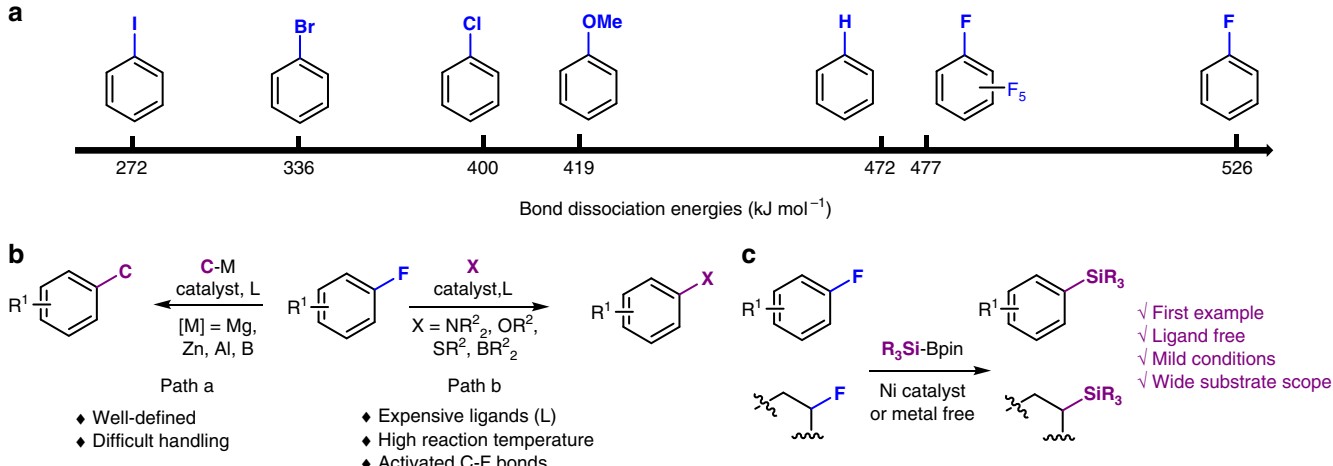

**Fig. 1** The bond strength and cross-coupling reaction via C–F cleavage. **a** Bond dissociation energies (BDEs); **b** Transition metal-catalyzed cross-coupling reactions for carbon–carbon and carbon–heteroatom bond-formations; **c** identification of defluorosilylation of unactivated fluoroarenes and alkyl fluorides under Ni catalyst or metal free

**Table 1 Optimization of the defluorosilylation process**

| Entry | Conditions | 3a (%)[a] | Entry | Conditions | 3a (%)[a] |
|---|---|---|---|---|---|
| 1 | Ni(COD)$_2$, KO$t$Bu, PhMe, 110 °C, 24 h | 56 | 7 | Ni(COD)$_2$, KOMe, $c$-hexane/THF 1/2 | 60 |
| 2 | Ni(COD)$_2$, KO$t$Bu, other solvents (e.g., $c$-hexane) | ≤74 | 8 | Ni(COD)$_2$, KHMDS, $c$-hexane/THF 1/2 | 56 |
| 3 | Ni(COD)$_2$, KO$t$Bu, $c$-hexane/THF 4/1 | 74 | 9 | Ni(PPh$_3$)$_2$Cl$_2$, KO$t$Bu, $c$-hexane/THF 1/2 | 20 |
| 4[b] | Ni(COD)$_2$, KO$t$Bu, $c$-hexane/THF 1/2 | 89 | 10 | Ni(acac)$_2$, KO$t$Bu, $c$-hexane/THF 1/2 | 12 |
| 5 | Ni(COD)$_2$, KO$t$Bu, $c$-hexane/THF 1/4 | 59 | 11[b] | none metal catalyst, KO$t$Bu, $c$-hexane/THF 1/2 | 0 |
| 6 | Ni(COD)$_2$, NaO$t$Bu, $c$-hexane/THF 1/2 | 71 | 12[b] | Ni(COD)$_2$, none base, $c$-hexane/THF 1/2 | 0 |

Reaction conducted on 0.2 mmol scale, 80 °C, 24 h
[a]Yield by gas chromatography. The reaction conditions (entry 4) are optimal
[b]rt, 2 h

of C–heteroatom bonds under certain conditions[14,15,19–21], such conditions did not work well for this transformation, even in the presence of additives such as CuF$_2$[22], CuI[14], LiCl[19], or TMAF[23] (Supplementary Table 2). We subsequently optimized the solvent system (entry 2, Table 1; Supplementary Table 3). Benzene and THF were equally effective as toluene for this reaction, while THF was better than the other solvents due to the suppression of hydrodefluorinated biphenyl as the main byproduct. Highly polar solvents such as 1,4-dioxane and DMF failed to provide **3a**. Changing the solvent to cyclohexane furnished **3a** in 74% yield. A binary solvent system (cyclohexane/THF = 1/2; v/v) afforded **3a** in 89% yield at room temperature (entry 4, Table 1). Other ratios of this solvent system did not improve the reaction outcome (entries 3–5, Table 1). Further optimizations of the solvent, base, and metal catalyst are provided in Supplementary Tables 3–5. A reduction of the catalyst loading to 5 mol% Ni(COD)$_2$ afforded a lower yield of **3a** (Supplementary Table 5). The base KO$t$Bu could be replaced with NaO$t$Bu, KOMe, or KHMDS, albeit under concomitant reduction of the yield (entries 6–8, Table 1). As shown in entries 9 and 10, other Ni catalysts resulted in a significant decrease of the yield of **3a**, similar to that observed with Cu-based catalysts (Supplementary Table 5). All reaction parameters were found to be critical for this in situ C–F bond cleavage with silylation reaction (entries 11 and 12, Table 1).

**Substrate scope**. With the optimal reaction conditions in hand (entry 4, Table 1), the scope of this ligand-free and mild Ni(COD)$_2$-catalyzed silylation of fluoroarenes **1** was investigated (Fig. 2a). A wide range of π-extended fluoroarenes (**1a**–**1l**), fluorophenyls (**1m**–**1w**), fluoropyridines (**1x**, **1aa**), and polyfluoroarenes (**1aa**–**1cc**) were efficiently converted into the corresponding aryl silanes (**3a**–**3cc**) using silylborane **2** as the coupling partner. Silylborane **2b** furnished **3b** in moderate yield under otherwise optimized reaction conditions. The chemoselectivity of this process is nicely illustrated by the fact that functional methoxy (**3c**, **3o**), dioxole (**3e**), silyl ether (**3t**), phenoxyl (**3q**), amine (**3r**), and boronic ester (**3v**) groups, which could be potentially cleaved in the presence of the Ni(0) complex, remained intact under the applied conditions[24–26]. The steric hindrance of *ortho*-substituents was also tolerated, as shown by the formation of aryl silanes **3j** and **3u**. It should be noted that fluoroarenes bearing pyrrole, thiophene, or indole rings also participated in the reaction, furnishing the desired defluoroarylsilanes **3f**, **3g**, **3h**, and **3k** in good yield, without observation of the expected sp$^2$ C–H bond activation–silylation products[27].

Notably, a series of substituted aryl silanes (**3m**–**3w**) were successfully prepared using reaction conditions identical to those for π-extended fluoroarenes, despite the different electronic state of the arenes, including high electron-donating NMe$_2$ substitution (**3r**). These results indicate that a nucleophilic aromatic substitution is highly unlikely to occur in this reaction. The reactivity of simple fluoroarenes was comparable to that of π-extended systems in this ligand-free and mild Ni-catalyzed defluorosilylation, rendering it a highly advantageous protocol for the preparation of aryl silanes. Particularly noteworthy is that fluoropyridine substrates (**1x**, **1aa**) can be transformed under these conditions. Substrates bearing more than one fluorine group (**1aa**–**1cc**) were also successfully monosilylated by carefully adjusting the stoichiometry of the reaction.

Prompted by our findings, we speculated that such a defluorosilylation strategy could also be extended to fluoroalkane substrates (sp$^3$ C–F silylation; Fig. 2b). Strikingly, with our system, a wide variety of alkyl fluorides were successfully converted into alkyl silanes in excellent yield, even in the absence of the Ni catalyst. Namely, alkyl fluorides including primary benzylic fluorides (**4a**–**4e**), secondary fluorides (**4f**–**4h**, **4j**), and an alkyl fluoride (**4l**) were smoothly transformed into the corresponding silylboranes using only KO$t$Bu as the activation agent. Defluorosilylation of fluorenyl fluoride **4i** and (1-fluorocyclobutyl)benzene (**4k**) was achieved using Ni(COD)$_2$ as the catalyst to provide alkyl silanes **5i** and **5k**, respectively. Remarkably, in the case of **4k** with tetra-substituted C–F bonds at the cyclobutyl ring, the silylation position was shifted to the neighboring secondary carbon to provide **5k**, presumably due to the steric hindrance. In all the cases, C–H bond silylation products were not detected[27].

**Synthetic application**. In light of these results, the major advantage of this mild and ligand-free C–F activation procedure is its tolerance toward a wide range of functional groups. We envisioned that our silylation protocol could be applied to the late-stage diversification of fluorine-containing bioactive molecules (Fig. 2c). As shown, steroid derivative **7** was synthesized in moderate yield (41%) under the same reaction conditions. Similarly, pitavastatin analogue **8**, bearing an electrophilic carbonyl group and an N-heteroaromatic ring, which in some cases may reduce the efficiency of metal catalysts, was converted into **9** in 50% yield. These results clearly demonstrate the potential utility of this defluorosilylation protocol as a platform for the late-stage functionalization of drugs. The silylated products may potentially serve as useful synthetic building blocks for versatile transformations, and some selected

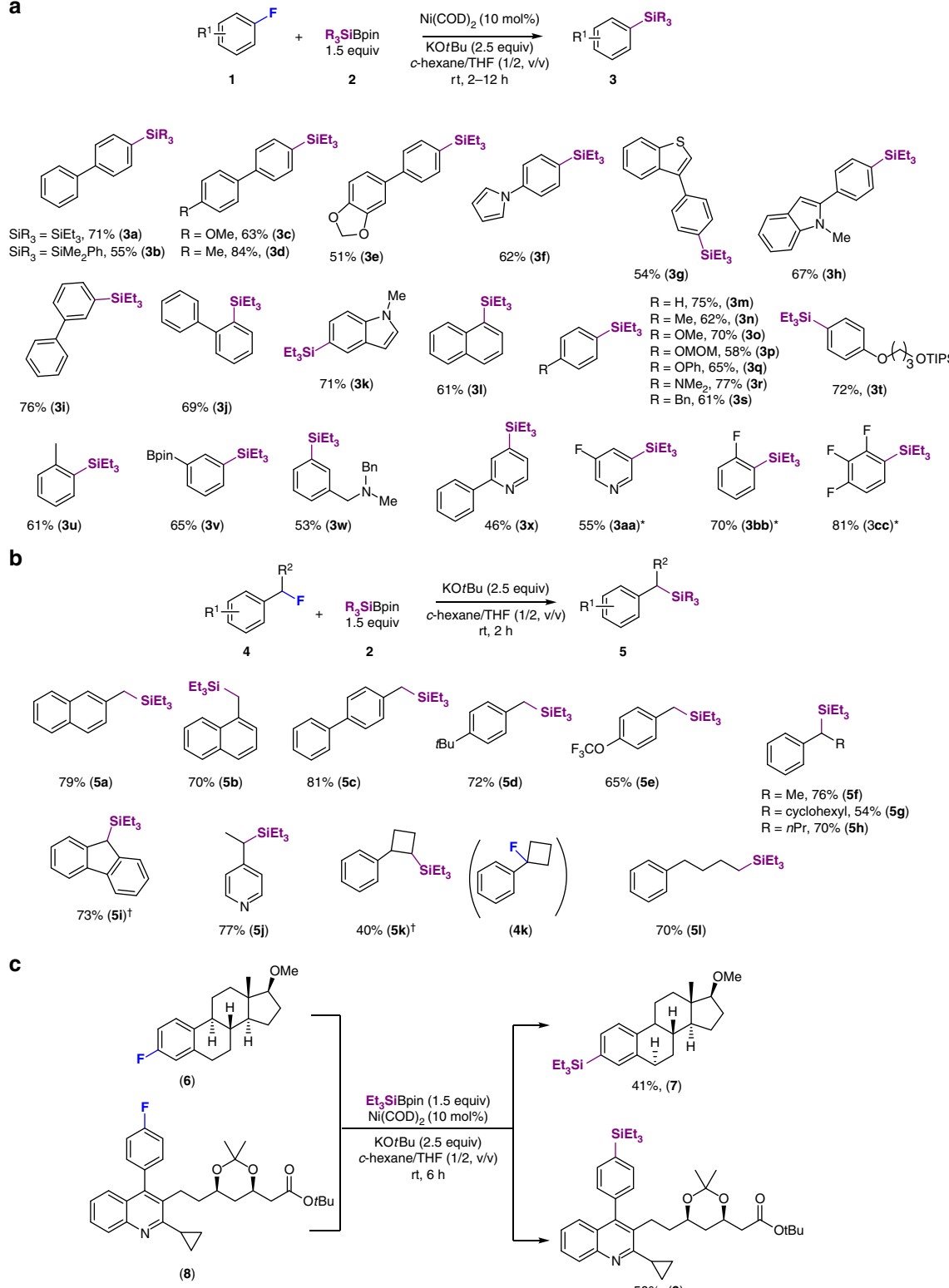

**Fig. 2** Scope of the defluorosilylation reactions. **a** Defluorosilylation of fluoroarenes. Standard reaction conditions: **1** (0.2 mmol, 1 equiv), R$_3$SiBpin (1.5 equiv), Ni(COD)$_2$ (10 mol%), KO$t$Bu (2.5 equiv), $c$-hexane/THF (1/2, v/v, 0.8 ml), rt, 2–12 h. Isolated yields. *Et$_3$Bpin (1.1 equiv). **b** Defluorosilylation of fluoroalkanes. Standard reaction conditions: **4** (0.2 mmol, 1 equiv), Et$_3$SiBpin (1.5 equiv), KO$t$Bu (2.5 equiv), $c$-hexane/THF (1/2, v/v, 0.8 ml), rt, 2 h. †Ni(COD)$_2$ (10 mol%). Isolated yields. **c** Late-stage defluorosilylation of fluorine-containing pharmaceuticals

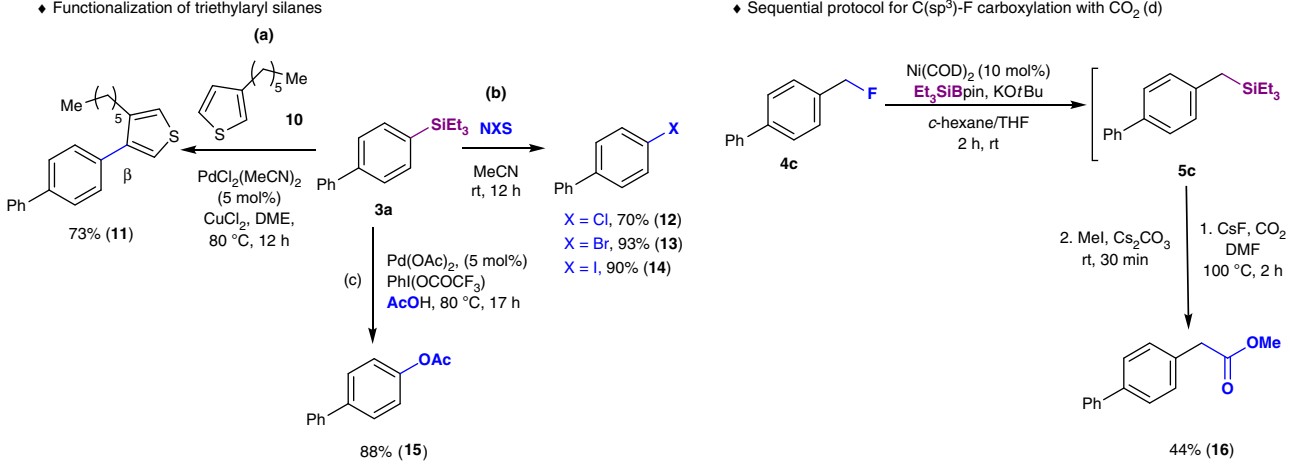

**Fig. 3** Synthetic application. **a** Pd-catalyzed C–H bond arylation of thiophene. **b** *ipso*-Halogenation. **c** Desilylative acetoxylation

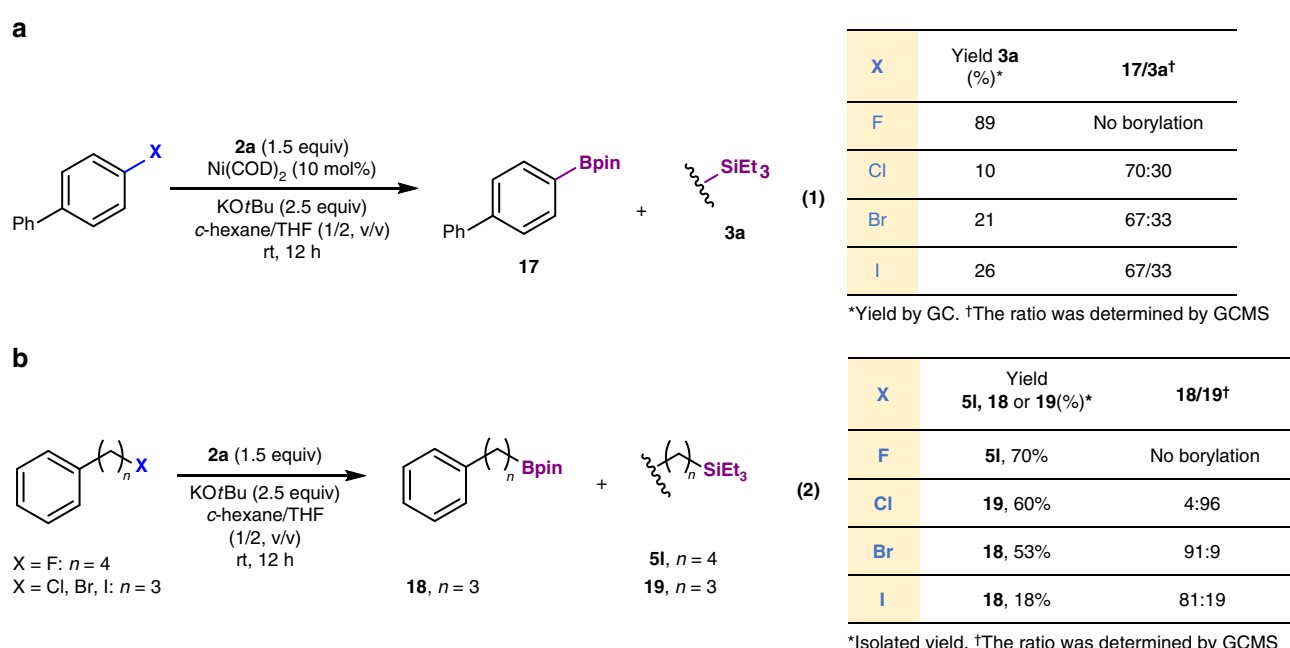

**Fig. 4** Competition between silylation and borylation of halogen-containing compounds. **a** Reaction of aryl halides (X = F, Cl, Br, I) and **2a**. **b** Reaction of alkyl halides (X = F, Cl, Br, I) and **2a**

synthetic applications of representative silanes obtained in this work are demonstrated (Fig. 3). For instance, triethylarylsilane **3a** can be readily coupled with thiophene via a Pd-catalyzed direct C–H arylation to give **11** in excellent yield with high β-selectivity (Fig. 3a)[28]. Furthermore, **3a** smoothly undergoes chlorination, bromination, and iodination to furnish **12**–**14** in good yield (Fig. 3b). The desilylative acetoxylation of **3a** was performed in the presence of Pd(OAc)$_2$ as a catalyst and PhI(OCOCF$_3$) in AcOH (Fig. 3c). This reaction represents a protocol for the phenolation of electron-rich aryl fluorides[29]. The catalytic carboxylation of the C($sp^3$)-F bond of **4c** was successfully achieved in moderate yield by the sequential reaction, i.e., via a Ni(COD)$_2$-catalyzed defluorosilylation, followed by a C($sp^3$)-Si bond activation to incorporate CO$_2$ under promotion of a fluoride anion (Fig. 3d)[30].

**Competition between silylation and borylation**. The use of silylboranes for the borylation of aryl, alkenyl, and alkyl halides

(Br, I) in the presence of an alkoxy base has been previously reported[31]. Interestingly, under the applied defluorosilylation conditions, chlorine-, bromine-, and iodine-substituted arenes and alkanes afforded a mixture of silylated and borylated compounds, whereby the borylated compounds were the major products, except for the alkyl chloride, which exhibited good selectivity toward the silylated product (Fig. 4). As has already been shown using fluoro-compounds (Fig. 2), borylation products are not detected using our protocol, while silanes are the main products via C–F bond functionalization. This behavior demonstrates the outstanding advantage of the use of fluorine in our reaction system compared to other halogens. Moreover, competitive experiments suggested that the C–F bond is more inert than the C–Cl, C–Br, and C–I bond in 1-chloro-3-fluorobenzene, 1-bromo-3-fluorobenzene, and 1-fluoro-3-iodobenzene under the conditions applied in this Ni(COD)$_2$-catalyzed defluorosilylation (Supplementary Fig. 3).

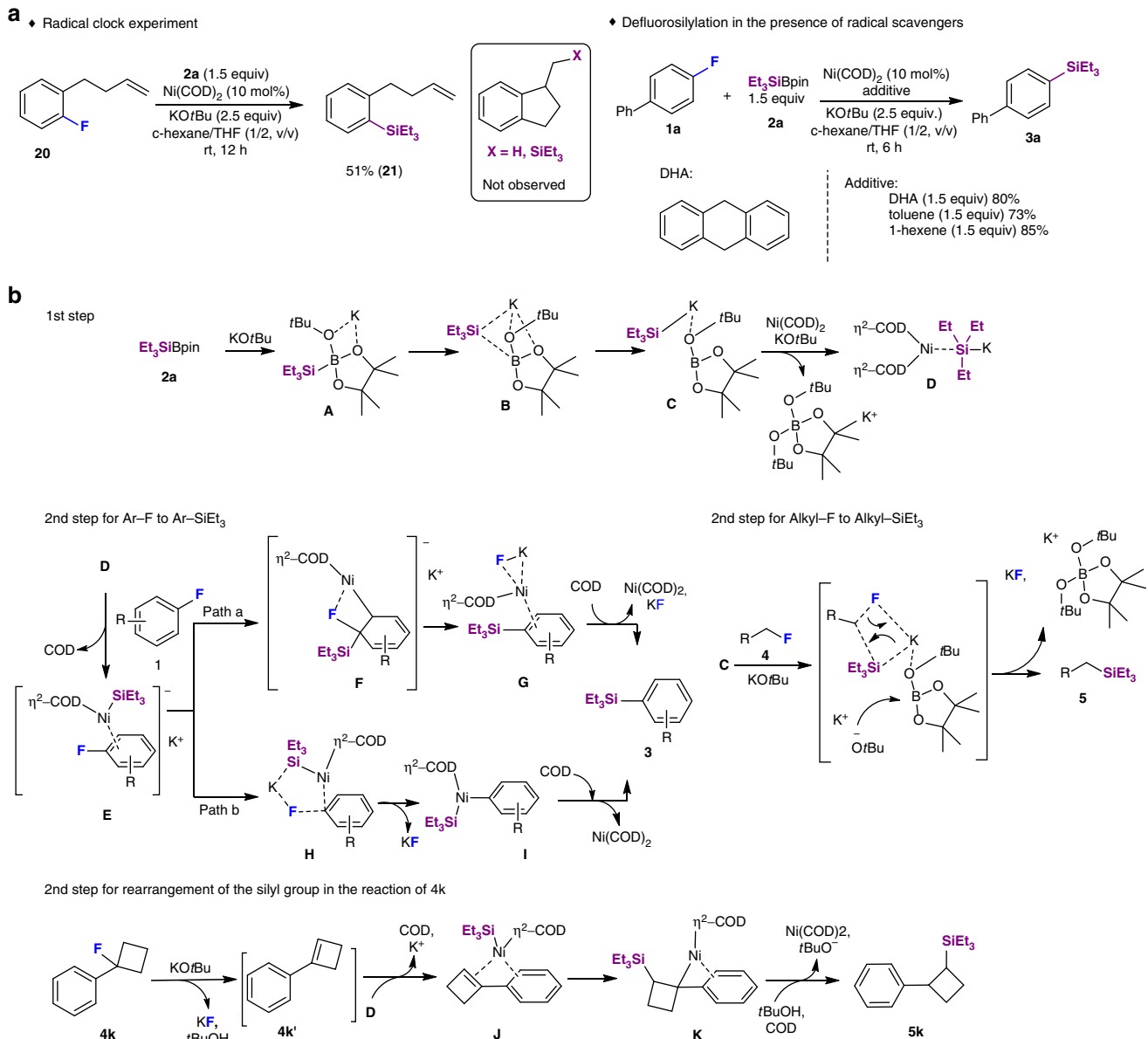

**Fig. 5** The studies of mechanism. **a** Investigation on radical-type pathways. **b** Proposed process for defluorosilylation of fluoroarenes and fluoroalkanes

**Mechanistic investigations**. Several experiments were conducted to gain insight into the mechanism of this transformation (Fig. 5a). To discuss the possibility of a radical-mediated reaction pathway, we attempted a radical-clock experiment using 4-(2-fluorophenyl)-1-butene (**20**) as the substrate. Under standard catalytic conditions, the reaction of **20** with Et₃SiBpin afforded defluorosilylated **21** (51%), while cyclization products were not detected (Fig. 5a). This result indicates that free radical species generated by single-electron transfer (SET) processes from the electron-rich Ni(0) center leading to carbon-centered radicals can be excluded[32]. In addition, we examined the reaction in the presence of radical scavengers, such as 9,10-dihydroanthracene (DHA), toluene, and 1-hexene (Fig. 5a). In all cases, the reactions proceeded smoothly to provide the desired **3a** in excellent yield, ruling out the generation of triethylsilyl radicals by either homolysis of the Si–B bond or from a silyl anion via SET[33]. Overall, these results suggest that the defluorosilylation reaction does not involve free radical intermediates.

**Proposed mechanism**. On the basis of the experimental results, we propose that the defluorosilylation reaction proceeds as shown in Fig. 5b. First, silylborane **2a** reacts with a mole of KOtBu to form intermediate **A**, which then rearranges into highly nucleophilic, silyl anionic species **C** via intermediate **B**[34]. Complex **C** undergoes a transmetalation with Ni(COD)₂ in the presence of a second mole of KOtBu to generate the active catalyst complex **D** under concomitant release of stable [BPin(OtBu)₂]K. The process from **2a** to **D**, i.e., the first step in Fig. 5b, has been accurately described by Avasare and co-workers based on DFT calculations[34]. We confirmed this reaction process by the detection of intermediate **A** by ¹¹B NMR studies (Supplementary Fig. 5). Species **D** generated in the first step undergoes a second step to furnish **3**. The mechanism from **D** with **1** to **3** has also been simulated based on the report by Avasare and co-workers. **D** reacts with fluoroarenes **1** to form π-nickel complex **E**, which may undergo two different pathways, i.e., a and b[34,35]. In pathway a, the silyl anion performs a nucleophilic attack on the fluorine-substituted carbon of the aryl ring by intramolecular SₙAr (I-

$S_NAr$), assisted by a transfer of the π-bond density of the aryl ring to the d-orbital of nickel (Fig. 5b, path a)[34,36,37]. The fluorine atom coordinates to the Ni center, where $K^+$ also assists the C–F bond cleavage to provide **G** via **F**. Finally, COD coordinates to the Ni center with the formed KF to provide aryl silanes **3**. The generation of KF was confirmed by $^{19}F$ NMR studies (Supplementary Figs. 6–8).

The ability of internal nucleophilic aromatic substitution (path a) is attributed to the $η^2$ coordination with nickel under retention of the aromaticity, which is more suitable for π-extended aromatic rings. However, simple fluoroarenes, for which the aromatic addition in path a is unfavorable, also successfully engaged in the defluorosilylation (Fig. 2), most likely due to relatively rigid ring and high barrier of aromaticity loss. Alternatively, C–F bond cleavage might occur through a non-classical oxidative pathway via the five-centered transition state **H**, followed by C–F bond cleavage. The C–F bond activation is assisted by the binding of $K^+$ to F to afford inert KF and the C-Ni intermediate **I**. The reductive elimination of **I** is assisted by COD to furnish **3** under concomitant regeneration of Ni(COD)$_2$. $^{19}F$ NMR experimental studies support the generation of KF (−213 ppm) and show two new peaks (−114 ppm and −135 ppm) for the potential intermediates such as **E**, **F**, **G**, and **H** during the transformation (Supplementary Fig. 6). The peak at −114 ppm appeared soon, while that for **1a** (−117 ppm) disappeared within 20 min. The peak at −114 ppm remained for 60 min and gradually disappeared over 20 h under concomitant emergence of a peak for KF (−213 ppm). Another peak at −135 ppm appeared after 40 min and disappeared after 60 min. Although more studies are necessary to fully interpret these results, transition states/charged intermediates such as $η^2$-arene nickel complexes **E**, **F**, **G**, **H**, and **I** may be involved in the activation of the C–F bond.

A conventional cross-coupling mechanism is also possible, in which C–F bond cleavage proceeds via an oxidative addition of aryl fluorides to Ni(0), followed by a transmetalation and a reductive elimination (Supplementary Fig. 9). Generally, the oxidative addition step needs high reaction temperature to overcome the C–F bond-cleavage barrier. However, our results indicate that the defluorosilylation could proceed under milder conditions (room temperature). Furthermore, electron-deficient substrates are normally preferred during the oxidative addition step. However, substrates such as trifluoromethyl- and pentafluorosulfanyl-substituted aryl fluorides did not engage in the defluorosilylation under the optimized reaction conditions (Supplementary Fig. 1). These results demonstrate that the C–F cleavage does not proceed via a conventional oxidative addition.

In the case of alkyl fluorides, the reaction does not depend on the presence of the Ni catalyst. The results indicate that the highly nucleophilic, silyl anionic species **C** directly reacts with alkyl fluorides **4** to **5** via an $S_N2$ process (Fig. 5b). Especially, the rearrangement of the silyl group in the reaction of **4k** is remarkable. To gain insight into the process of this transformation, we have examined the reaction of **4k** in the absence/presence of **2a** and without Ni(COD)$_2$. These reactions afforded only aryl cyclobutene **4k′** via an elimination pathway (Supplementary Fig. 10, eq1). However, **4k** could be efficiently converted into **5k** in the presence of Ni(COD)$_2$ (Supplementary Fig. 10, eq2), which indicates that **4k′** serves as an intermediate for the defluorosilylation of **4k**. Based on this notion, we have developed a hypothetical mechanism (Fig. 5b). Initially, the elimination of the fluorine from the cyclobutyl ring is triggered by KOtBu to give **4k′**, followed by coordinating of the silane/Ni(COD)$_2$ base complex **D** to generate the intermediate $η^2$-arene nickel complex **J**[38,39]. Subsequently, **J** is subject to an addition and reductive elimination to afford the corresponding silane (**5k**).

## Discussion

We have presented an efficient Ni-based catalytic system that is capable of activating the C–F bonds of fluoroarenes in the absence of additional ligands to provide aryl silanes in good to high yield. This protocol for the preparation of aryl silanes is characterized by its mild conditions and wide substrate scope of inert fluoroarenes. The defluorosilylation of alkyl fluorides is directly achieved under very similar conditions, even in the absence of a Ni catalyst. Given the widespread presence of fluorinated compounds in commercially available chemicals, drugs, and functional materials, and the wide range of applications of organosilicon compounds, this strategy should become a powerful tool in organic synthesis, especially for the structural design and potential late-stage functionalization of drug lead compounds.

## Methods

**General procedure for defluorosilylation of aryl fluorides**. To a flame-dried screw-capped test tube were sequentially added a fluoroarene **1** (0.20 mmol, 1 equiv), silylborane **2** (0.30 mmol, 1.5 equiv), Ni(COD)$_2$ (5.5 mg, 0.02 mmol, 10 mol %), KOtBu (56.1 mg, 0.5 mmol, 2.5 equiv) and c-hexane/THF (1/2, v/v, 0.8 ml) in a glovebox filled with argon gas. The tube with the mixture was sealed and removed from the glovebox, and stirred at room temperature for 2–12 h. The reaction progress was monitored by TLC. Then, the mixture was added saturated aqueous ammonium chloride (3 ml) and extracted with EtOAc (3 × 3 ml). The combined organic extract was dried over MgSO$_4$ and filtrated. The filtrate was concentrated under reduced pressure. The residue was purified to give the silanes.

**General procedure for defluorosilylation of alkyl fluorides**. To a flame-dried screw-capped test tube were sequentially added a fluoroalkanes **1** (0.20 mmol, 1 equiv), silylborane **2a** (0.30 mmol, 1.5 equiv), KOtBu (56.1 mg, 0.5 mmol, 2.5 equiv) and c-hexane/THF (1/2, v/v, 0.8 ml) in a glovebox filled with argon gas. The tube with the mixture was sealed and removed from the glovebox, and stirred at room temperature for 2–12 h. The reaction progress was monitored by TLC. The mixture was then added saturated aqueous ammonium chloride (3 ml) and extracted with EtOAc (3 × 3 ml). The combined organic extract was dried over MgSO4 and filtrated. The filtrate was concentrated under reduced pressure. The residue was purified to give the silanes.

## Data availability

The data supporting the findings of this study are available within the paper and its Supplementary Information. All relevant data are also available from the authors.

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

## Acknowledgements

This work was supported by JSPS KAKENHI grants JP 18H02553 (KIBAN B) and JP 18H04401 (Middle Molecular Strategy), as well as the Asahi Glass Foundation. B.C. thanks CSC for CSC Scholarship.

## Author contributions

N.S. conceived the concept. B.C. optimized the reaction conditions and surveyed the substrate scope. S.J. prepared the starting materials. N.S. directed the project. N.S., E.T., and B.C. prepared the manuscript.

## Additional information

**Competing interests:** The authors declare no competing interests.

