## [Peer Review File · Nature Communications]

Reviewer #1 (Remarks to the Author):

In this paper, Shibata and co-workers report the transformation of aryl and alkyl fluorides into the corresponding silanes (triethylsilyl derivatives) through a defluorosilylation process under mild reaction conditions (room temperature, 2-12 h). The reactions are performed using Et₃SiB(pinacolato) as the silylating reagent, in the presence of 10 mol% of Ni(COD)₂ and 2.5 equivalents of a strong base (KOt-Bu), in a mixture of cyclohexane and THF, under argon. Surprisingly, the inert and strong C-F bond is involved in these reactions. It is worth mentioning that the methodology here presented exhibits broad substrate scope (aromatic, heteroaromatic, benzylic, secondary and primary alkyl fluorides) and high functional group tolerance. Importantly, structures of unsuccessful substrates for this nickel-catalyzed defluorosilylation are reported in Supplementary Information. A plausible mechanism to explain the defluorosilylation of aryl fluorides has been proposed. However, the reaction proceeds even in the absence of the nickel catalyst for alkyl fluorides, so a different mechanism should be involved for these substrates (the authors proposed a S_N2 process). On the other hand, in order to explain the formation of compound 5k, instead of a S_N2 process, formation of a phenonium ion intermediate would explain better the regiochemistry. Another interesting finding is the formation of the borylated product as the major component of the reaction mixture, instead of the silyl derivative, when this protocol is applied to the corresponding aryl (biphenyl) or alkyl (3-phenylpropyl) bromide. This demonstrates the utility of fluorine derivatives in these transformations.

I found this study of interest for synthetic organic chemists, and therefore recommend the publication of this paper.

Corrections to be made and some remarks:

1) Some re-reading should be made to remove the typos and other errors.

1a) Page 1, left column, 1st paragraph, line 1.- "The directed activation of..." should be "Direct activation of..."?

1b) Page 1, left column, 1st paragraph, line 15.- "Here we demonstrate that a rare..." should be "Here we demonstrate a rare..."?

1c) Page 1, left column, 2nd paragraph, line 1.- "In organic compounds that contain a carbon-fluorine..." should be "Organic compounds that contain a carbon-fluorine..."

.....

2) Please, check carefully References, in particular dots, commas, capital letters, etc. For instance:

2a) Reference 2.- "Ahrens, T., Kohlmann, J., Ahrens, M., Braun, T." should be "Ahrens, T., Kohlmann, J., Ahrens, M. & Braun, T."? (all references)

2b) Reference 4.- "kirsch, P. (ed.)..." should be "Kirsch, P. (ed.)..."

2c) Reference 6.- "... (2011)." should be "... (2011)."

.....

3. Concerning the chemoselectivity of C-halogen activation in these nickel-catalyzed reactions, did the authors study the reaction of a bromofluoroarene (1-bromo-3-fluorobenzene, for instance) in order to know the order of the reactivity of the C-Br and C-F bonds?

Reviewer #2 (Remarks to the Author):

In this manuscript is described nickel-catalyzed ipso-silylation of fluoroarenes that involve cleavage of a chemically stable C–F bond. As the authors discussed in the introduction part, derivatization via C–F bond cleavage has potential utility for preparation of valuable compounds, while the cleavage is quite difficult task. For this purpose, several groups have achieved defluoroborylation of fluoroarenes (refs 12 and 13), whereas the corresponding silylation has been firstly developed in this work. Comparing to the reported borylation method, the silylation shows significant mild conditions and moderate substrate scope. Study for optimization was also well done. Moreover, expansion of the silylation into aliphatic fluorides is remarkable.

I think one of the most critical issue on this work is utility of the silylated products. While the authors argued synthetic utility of organosilane compounds, it is only the case when the silyl group has enough Lewis acidity to be silicate under basic conditions, such as a hydrosilyl group or a alkoxy-silyl group. In this manuscript, the authors show introduction of a SiEt₃ and a SiMe₂Ph group, which have only a weak Lewis acidity. Therefore, I'm afraid that the arylsilanes afforded by this method could not be utilized for further derivatization. To avoid such possibility, the authors should demonstrate derivatization of the products, which will improve the value of this manuscript. In addition, the product of the silylation of alkyl fluorides is alkylsilanes, which is likely to be used for derivatization. Additional discussion to clarify value of this transformation would be required in the main text.

Another small issue is mechanistic consideration. The authors discussed the reaction mechanism almost based on the literature for theoretical analysis of nickel-catalyzed silylation of aryl methyl ethers (ref 28). The authors have also tried to observe some byproducts by NMR technique to confirm the mechanism, but the experimental results they showed in this manuscript are very few. Although I imagine that experiments to gain insight into the mechanism more deeply, such as isolation of intermediates or kinetics measurement, is difficult to conduct, I would ask the authors to add more detailed discussion from the data shown in the manuscript. For example, the authors described two pathways to afford the silylarenes (paths a and b in Figure 5). Which is more plausible? In the study of substrate scope, the authors found that reactivity of simple fluoroarenes are comparable to that of π -extended substrates such as biaryls, which suggested that path b is likely to be plausible because this path does not involve the loss of aromaticity. In addition, the author

firstly should address the possible conventional mechanism involving oxidative addition of the C–F bond into Ni(0) species followed by transmetalation and reductive elimination, such as Martin's work (ref 13). Additional comment for this possibility should be preferred. Since new mechanistic concept was not so much described in this manuscript, additional discussion based on the authors' results is desired to clarify mechanistic novelty.

Overall, I think this manuscript will become to deserve publication in the Nature Communications after revision on the points addressed above as well as listed below.

Selected points

- The authors prepared silylboranes by using an iridium catalyst. Have the authors confirmed contamination of the iridium complex in the silylboranes? Possible effect of the trace amount of the iridium to the defluorosilylation reactions should be considered. For example, the silylation in the presence of iridium complexes should be conducted to remove the possibility.
- Whereas the authors tested a broad range of substrates and unsuccessful list (Page S11), a substrate bearing acidic proton, such as amides or alcohols, could not be found. To clarify synthetic utility of this method, I would ask the authors to test such substrates.
- Rearrangement of the silyl group in the reaction of 4k is remarkable result. I would recommend the authors to add a plausible mechanism of the rearrangement.
- As shown in the Figure 4, the effect of the halogens seems interesting. To clarify the effect, additional experiments for the reaction using aryl iodide or chloride is desired.

Minor points

- Page 3, 3rd paragraph, line 5: A typo (2-(3-tutenyl)-...) is found.
- Page 3, 3rd paragraph, line 13: DHA would be 9,10-dihydroanthracene. In addition, Galvinoxyl free radical should also be tested.
- Page 3, 6th paragraph, line 10: The intermediate "I" would be a mistake of "E". Please reconsider.
- Page S11: Add a value for ^{11}B NMR shift of Et_3SiBpin .
- Some HRMS-data shown in the Supplementary Information is not acceptable. Basically, difference between a calculated m/z value and a measured one must be in approximately 5 ppm. However, I found big differences in the data for 4e, 3e, 3f, 3k, 3n, 3o, 3cc, 5d-j, 5l, S7, 8, 9, and 16. Some of other data give also relatively big differences in the data. Since these data does not make sense for characterization, the authors should recheck the HRMS-data and remeasure them if needed.

**Our point-by-point responses to the reviewers' requests for
"Defluorosilylation of Fluoroarenes and Fluoroalkanes"**

Reviewer #1:

However, the reaction proceeds even in the absence of the nickel catalyst for alkyl fluorides, so a different mechanism should be involved for these substrates (the authors proposed a SN2 process). On the other, hand, in order to explain the formation of compound 5k, instead of a SN2 process, formation of a phenonium ion intermediate would explain better the regiochemistry.

Response: Thank you very much for your comments on the reaction mechanism for the defluorosilylation of alkyl fluorides. We have further examined the reaction of **4k** in the absence/presence of **2a** and in the absence/presence of Ni(COD)₂. These reactions afforded only aryl cyclobutene **4k'** via an elimination reaction (Fig. 1, eq. 1). However, **4k** could be efficiently converted into **5k** in the presence of Ni(COD)₂ (Fig. 1, eq. 2), which suggests that **4k'** serves as an intermediate for the defluorosilylation of **4k**. The phenonium ion is a classical and familiar intermediate in cationic rearrangement reactions. In case of the **4k** building block, a cation would need to be furnished initially at the β position of the cyclobutyl ring to close the arene. It is, however, unlikely that a cation at the β position is formed under the applied reaction conditions. Therefore, we have considered a modified hypothetical mechanism (*cf.* Fig. 2).

We have added a short comment in the revised manuscript:

Page 4, right column, 3rd paragraph, line 5:

Especially, the rearrangement of the silyl group in the reaction of **4k** is remarkable. To gain insight into the process of this transformation, we have examined the reaction of **4k** in the absence/presence of **2a** and in the absence/presence of Ni(COD)₂. These reactions afforded only aryl cyclobutene **4k'** via an elimination pathway (Supplementary Fig. S8, eq1). However, **4k** could be efficiently converted into **5k** in the presence of Ni(COD)₂ (Supplementary Fig. S8, eq2), which indicates that **4k'** serves as an intermediate for the defluorosilylation of **4k**. Based on this notion, we have developed a hypothetical mechanism (Fig. 5B). Initially, the elimination of the fluorine from the cyclobutyl ring is triggered by KOtBu to give **4k'**, followed by coordinating of the silane/Ni(COD)₂ base complex **D** to generate the intermediate η²-arene nickel complex **J**.^{38, 39} Subsequently, **J** is subject to an addition and reductive elimination to afford the corresponding silane (**5k**).

Fig. 1. (Fig. S8) Examining the defluorosilylation step in the reaction of **4k**.

(Corresponding figure number in the Supplementary Information is given in the parentheses)

Fig. 2. (Fig. 5B) Hypothetical mechanism for the defluorosilylation of **4k**.

(Corresponding figure number in the text is given in the parentheses)

Question 1: *Page 1, left column, 1st paragraph, line 1.- “The directed activation of...” should be “Direct activation of...”*

Response: We have altered this sentence in the revised version to: *Page 1, left column, abstract, line 1:* “Direct activation of carbon-fluorine bonds (C-F) to introduce ...”

Question 2: *Page 1, left column, 1st paragraph, line 15.- “Here we demonstrate that a rare...” should be “Here we demonstrate a rare...”*

Response: We have changed this sentence to “Here, we demonstrate an unprecedented ipso-silylation of aryl fluorides via cleavage of unactivated C–F bonds by a Ni catalyst under mild conditions and without the addition of any external ligand” in the revised version, *Page 1, left column, abstract, line 9.*

Question 3: *Page 1, left column, 2nd paragraph, line 1.- “In organic compounds that contain a carbon-fluorine...” should be “Organic compounds that contain a carbon-fluorine...”*

Response: We have altered this sentence to “Organic compounds that contain a carbon-fluorine...” in *Page 2, left column, 1st paragraph, line 1.*

Question 4: *Please, check carefully References, in particular dots, commas, capital letters, etc. For instance: 2a) Reference 2.- “Ahrens, T., Kohlmann, J., Ahrens, M., Braun, T.” should be “Ahrens, T., Kohlmann, J., Ahrens, M. & Braun, T.”? (all references). 2b) Reference 4.- “kirsch, P. (ed.)...” should be “Kirsch, P. (ed.)...”. 2c) Reference 6.- “...(2011).” should be “...(2011).”*

Response: We have revised all references according to the *Nature Communications* style in the revised version.

Question 5: *Concerning the chemoselectivity of C-halogen activation in these nickel-catalyzed reactions, did the authors study the reaction of a bromofluoroarene (1-bromo-3-fluorobenzene, for instance) in order to know the order of the reactivity of the C-Br and C-F bonds?*

Response: Thank you so much for your comments on the reactivity of the C-Br and C-F bonds. We have examined the chemoselectivity of C-halogen activation in these nickel-catalyzed reactions (Fig. 3). In brief: the C-F bond is more stable than the C-Cl, C-Br and C-I bonds in 1-chloro-3-fluorobenzene, 1-bromo-3-fluorobenzene, and 1-fluoro-3-iodobenzene under the standard conditions, which affords triethyl(3-fluorophenyl)silane and 2-(3-fluorophenyl)-4,4,5,5-tetramethyl-1,3,2-dioxaborolane as the major products with trace amounts of 1,3-bis(triethylsilyl)benzene and

triethyl(3-(4,4,5,5-tetramethyl-1,3,2-dioxaborolan-2-yl)phenyl)silane. We have included these results in the Supplementary Information and added a short comment in the discussion section of the revised manuscript:

Page 3, right column, 2nd paragraph, line 15:

Moreover, competitive experiments suggested that the C-F bond is more inert than the C-Cl, C-Br, and C-I bond in 1-chloro-3-fluorobenzene, 1-bromo-3-fluorobenzene, and 1-fluoro-3-iodobenzene under the conditions applied in this Ni(COD)₂-catalyzed defluorosilylation (Supplementary Fig. S2).

Fig. 3. (Fig. S2) The reactivity of the C-F bond compared the C-Cl, C-Br, and C-I bond, in 1,3-disubstituted benzene derivatives.

(Corresponding figure number in the Supplementary Information is given in the parentheses)

Reviewer #2:

I think one of the most critical issue on this work is utility of the silylated products. While the authors argued synthetic utility of organosilane compounds, it is only the case when the silyl group has enough Lewis acidity to be silicate under basic conditions, such as a hydrosilyl group or a alkoxy-silyl group. In this manuscript, the authors show introduction of a SiEt₃ and a SiMe₂Ph group, which have only a weak Lewis acidity. Therefore, I'm afraid that the arylsilanes afforded by this method could not be utilized for further derivatization. To avoid such possibility, the authors should demonstrate derivatization of the products, which will improve the value of this manuscript. In addition, the product of the silylation of alkyl fluorides is alkylsilanes, which is likely to be used for derivatization. Additional discussion to clarify value of this transformation would be required in the main text.

Response: We have examined the potential further derivatization of arylsilanes and alkylsilanes in Fig. 4. We have added a short comment to the discussion section in the revised manuscript:

Page 3, right column, 1st paragraph, Line 14:

The silylated products may potentially serve as useful synthetic building blocks for versatile transformations, and some selected synthetic applications of representative silanes obtained in this work are demonstrated (Fig. 3). For instance, triethylarylsilane **3a** can be readily coupled with thiophene via a Pd-catalyzed direct C-H arylation to give **11** in excellent yield with high β -selectivity (Fig. 3a).²⁸ Furthermore, **3a** smoothly undergoes chlorination, bromination, and iodination to furnish **12-14** in good yield (Fig. 3b). The desilylative acetoxylation of **3a** was performed in the presence of Pd(OAc)₂ as a catalyst and PhI(OCOCF₃) in AcOH (Fig. 3c). This reaction represents a protocol for the phenolation of electron-rich aryl fluorides.²⁹ The catalytic carboxylation of the C(sp³)-F bond of **4c** was successfully achieved in moderate yield by the sequential reaction, i.e., via a Ni(COD)₂-catalyzed

defluorosilylation, followed by a C(sp³)-Si bond activation to incorporate CO₂ under promotion of a fluoride anion (Fig. 3d).³⁰

Fig. 4. (Fig. 3) Selected synthetic applications of representative silanes obtained in this work. (Corresponding figure number in the text is given in the parentheses)

Another small issue is mechanistic consideration. The authors discussed the reaction mechanism almost based on the literature for theoretical analysis of nickel-catalyzed silylation of aryl methyl ethers (ref 28). The authors have also tried to observe some byproducts by NMR technique to confirm the mechanism, but the experimental results they showed in this manuscript are very few. Although I imagine that experiments to gain insight into the mechanism more deeply, such as isolation of intermediates of kinetics measurement, is difficult to conduct, I would ask the authors to add more detailed discussion from the data shown in the manuscript. For example, the authors described two pathways to afford the silylarenes (paths a and b in Figure 5). Which is more plausible? In the study of substrate scope, the authors found that reactivity of simple fluoroarenes are comparable to that of π -extended substrates such as biaryls, which suggested that path b is likely to be plausible, because this path does not involve the loss of aromaticity. In addition, the author firstly should address the possible conventional mechanism involving oxidative addition of the C–F bond into Ni(0) species followed by transmetalation and reductive elimination, such as Martin's work (ref 13). Additional comment for this possibility should be preferred. Since new mechanistic concept was not so much described in this manuscript, additional discussion based on the authors' results is desired to clarify mechanistic novelty.

Response: We agree with your comments. In this study, we have carried out several experiments to gain insight into the potential process of this defluorosilylation, including radical-clock experiments to exclude free radical species generated by SET processes from the electron-rich Ni(0) center leading to carbon-centered radicals, and radical scavenger tests to rule out the generation of triethylsilyl radicals by either hemolysis of the Si-B bond or from a silyl anion via SET. Moreover, as aryl fluorides **3n-3t** were successfully converted into aryl silanes, this rules out the possibility of a nucleophilic aromatic substitution process. Based on our experimental data and literature research (*Organometallics* **2018**, 37, 1141-1149; *J. Am. Chem. Soc.* **2017**, 139, 1191-1197; *Chem.-Eur. J.* **2017**, 23, 17249-17256; *Organometallics* **2016**, 35, 2053-2056), we favor the two fundamental mechanistic pathways: internal nucleophilic substitution and non-classical oxidative addition. This defluorosilylation of the aryl fluoride is rapidly complete,

and it is thus difficult to isolate intermediates due to their sensitivity and short lifetime. Although more kinetic experiments are necessary to fully interpret these results, charged intermediates/transition states, such as η^2 -arene nickel complexes **E**, **F**, **G**, **H** and **I** may be involved in the activation of the C-F bond.

We agree that path b (non-classical oxidative addition) is more plausible for the transformation of simple aryl fluorides.

We have added new comments and a conventional cross-coupling mechanism to interpret the mechanism in the discussion:

Page 4, left column, 3rd paragraph, line 1:

The ability of internal nucleophilic aromatic substitution (path a) is attributed to the η^2 coordination with nickel under retention of the aromaticity, which is more suitable for π -extended aromatic rings. However, simple fluoroarenes, for which the aromatic addition in path a is unfavorable, also successfully engaged in the defluorosilylation (Fig. 2), most likely due to relatively rigid ring and high barrier of aromaticity loss.

Page 4, right column, 1st paragraph, line 5:

Although more studies are necessary to fully interpret these results, transition states/charged intermediates such as η^2 -arene nickel complexes **E**, **F**, **G**, **H** and **I** may be involved in the activation of the C-F bond.

Page 4, right column, 2nd paragraph, line 1:

A conventional cross-coupling mechanism is also possible, in which C-F bond cleavage proceeds via an oxidative addition of aryl fluorides to Ni(0), followed by a transmetalation and a reductive elimination (Supplementary Fig. S7). Generally, the oxidative addition step needs high reaction temperature to overcome the C-F bond cleavage barrier. However, our results indicate that the defluorosilylation could proceed under milder conditions (room temperature). Furthermore, electron-deficient substrates are normally preferred during the oxidative addition step. However, substrates such as trifluoromethyl- and pentafluorosulfonyl-substituted aryl fluorides did not engage in the defluorosilylation under the optimized reaction conditions (Supplementary Fig. S1). These results demonstrate that the C-F cleavage does not proceed via a conventional oxidative addition.

We have added a scheme of the conventional cross-coupling mechanism for Ni-catalyzed defluorosilylations to the Supplementary Information:

Fig. 5. (**Fig. S7**) Proposed a conventional cross-coupling mechanism in defluorosilylation reaction.

(Corresponding figure number in the Supplementary Information is given in the parentheses)

Question 1: *The authors prepared silylboranes by using an iridium catalyst. Have the authors confirmed contamination of the iridium complex in the silylboranes? Possible effect of the trace amount of the iridium to the defluorosilylation reactions should be considered. For example, the silylation in the presence of iridium complexes should be conducted to remove the possibility.*

Response: The defluorosilylation of 4-fluoro-1,1'-biphenyl in the presence of the iridium complex $[\text{Ir}(\text{COD})\text{OMe}]_2$ (Fig. 6) affords the corresponding defluorosilylated product in low yield together with dehydrosilylated compounds as by-products (detected by GCMS). Such iridium complexes are thus not efficient catalysts for the formation of aryl silanes by C-F cleavage. On the other hand, 2-(dimethylphenylsilyl)-4,4,5,5-tetramethyl-1,3,2-dioxaborolane (**2b**), which is commonly prepared by the treatment of dimethylphenylsilyllithium with pinacolborane in the absence of iridium-based catalysts (*Organometallics* **2000**, *19*, 4647-4649), furnished **3b** in moderate yield under the optimized defluorosilylation conditions. Furthermore, we washed a solution of silylborane (**2a**) in EtOAc with water in order to remove any potentially contaminating metal ions. Subsequently, the solution was dried over anhydrous MgSO_4 , filtered, and concentrated under reduced pressure. The thus obtained **2a** was treated with **1a** under the same optimized reaction conditions to afford the corresponding product in comparable yield.

Fig. 6

Question 2: *Whereas the authors tested a broad range of substrates and unsuccessful list (Page S11), a substrate bearing acidic proton, such as amides or alcohols, could not be found. To clarify synthetic utility of this method, I would ask the authors to test such substrates.*

Response: We have already examined substrates with acidic protons (Fig. 7). Substrates bearing amide or amine substituents are not suitable for these reactions, despite increasing the amount of base (one equivalent of $\text{KO}t\text{Bu}$) and increasing the reaction temperature (80 °C). Moreover, substrates with hydroxy substituents afforded only trace amounts of the defluorosilylated product (GCMS).

Fig. 7

Question 3: *Rearrangement of the silyl group in the reaction of 4k is remarkable result. I would recommend the authors to add a plausible mechanism of the rearrangement.*

Response:

Thank you for your comment on the reaction mechanism for the defluorosilylation of alkyl fluorides. We have further examined the reaction of **4k** in the presence/absence of **2a** and in the presence/absence of Ni(COD)_2 . The only product that we were able to detect was aryl cyclobutene **4k'** through an elimination reaction (Fig. 1, eq. 1). However, **4k** could be efficiently converted into **5k** in the presence of Ni(COD)_2 (Fig. 1, eq. 2), which indicates that **4k'** is an intermediate in the defluorosilylation of **4k**. So we consider a hypothetical mechanism is as follows (Fig. 2).

We have added a short comment in reviewed version:

Page 4, right column, 3rd paragraph, line 5:

Especially, the rearrangement of the silyl group in the reaction of **4k** is remarkable. To gain insight into the process of this transformation, we have examined the reaction of **4k** in the absence/presence of **2a** and in the absence/presence of Ni(COD)_2 . These reactions afforded only aryl cyclobutene **4k'** via an elimination pathway (Supplementary Fig. S8, eq1). However, **4k** could be efficiently converted into **5k** in the presence of Ni(COD)_2 (Supplementary Fig. S8, eq2), which indicates that **4k'** serves as an intermediate for the defluorosilylation of **4k**. Based on this notion, we have developed a hypothetical mechanism (Fig. 5B). Initially, the elimination of the fluorine from the cyclobutyl ring is triggered by KOtBu to give **4k'**, followed by coordinating of the silane/ Ni(COD)_2 base complex **D** to generate the intermediate η^2 -arene nickel complex **J**.^{38, 39} Subsequently, **J** is subject to an addition and reductive elimination to afford the corresponding silane (**5k**).

Fig. 1. (Fig. S8) Examining the defluorosilylation step in the reaction of **4k**.
(Corresponding figure number in the Supplementary Information is given in the parentheses)

Fig. 2. (Fig. 5B) Hypothetical mechanism for the defluorosilylation of **4k**.
(Corresponding figure number in the text is given in the parentheses)

Question 4: As shown in the Figure 4, the effect of the halogens seems interesting. To clarify the effect, additional experiments for the reaction using aryl iodide or chloride is desired.

Response: We have already examined the influence of the halogens (fluorine, chlorine, bromine, and iodine) on our defluorosilylation protocol (Fig. 8).

We have added new comments to the discussion section of the revised manuscript:

Page 3, right column, 2nd paragraph, line 4.

Interestingly, under the applied defluorosilylation conditions, chlorine-, bromine-, and iodine-substituted arenes and alkanes afforded a mixture of silylated and borylated compounds, whereby the borylated compounds were the major products, except for the alkyl chloride which exhibited good selectivity toward the silylated product.

Fig. 8. (Fig. 4) Competition between silylation and borylation of halogen-containing compounds
(Corresponding figure number in the text is given in the parentheses)

Question 5: Page 3, 3rd paragraph, line 5: A typo (2-(3-tutenyl)-...) is found.

Response: We have revised (2-(3-tutenyl)-fluorobenzene into “4-(2-fluorophenyl)-1-butene” in Page 3, right column, 3rd paragraph, line 5.

Question 6: Page 3, 3rd paragraph, line 13: DHA would be 9,10-dihydroanthracene. In addition, Galvinoxyl free radical should also be tested.

Response: We have altered docosahexaenoic acid to 9,10-dihydroanthracene on *Page 4, left column, 1st paragraph, line 4.*

Galvinoxyl free radicals would not be suitable for this Ni(COD)₂-catalyzed defluorosilylation system, as the silane/Ni(COD)₂ base complex that is formed quickly by treatment of silylborane **2a** with KO^tBu in the presence of Ni(COD)₂ could react with the Galvinoxyl free radical followed by a reductive elimination to give Galvinoxyl silane.

Question 7: *Page 3, 6th paragraph, line 10: The intermediate "I" would be a mistake of "E". Please reconsider.*

Response: We have revised this sentence into "...intermediates such as **E**, **F**, **G** and **H** during the transformation." on *Page 4, 3rd paragraph, line 17.*

Question 8: *Page S11: Add a value for 11B NMR shift of Et3SiBpin.*

Response: The ¹¹B NMR data of Et₃SiBpin have been added on *Page S47* in Supplementary Information.

¹¹B NMR (225 MHz, CDCl₃) δ 34.24.

Question 9: *Some HRMS-data shown in the Supplementary Information is not acceptable. Basically, difference between a calculated m/z value and a measured one must be in approximately 5 ppm. However, I found big differences in the data for 4e, 3e, 3f, 3k, 3n, 3o, 3cc, 5d-j, 5l, S7, 8, 9, and 16. Some of other data give also relatively big differences in the data. Since these data does not make sense for characterization, the authors should recheck the HRMS-data and remeasure them if needed.*

Response: We have rechecked all HRMS data to make sure the accuracy is within approximately 5 ppm:

4e: HRMS (EI) [C₈H₆F₄O] (M⁺) *calcd.* 194.0355, *found* 194.0352.
3e: HRMS (EI) [C₁₉H₂₄O₂Si] (M⁺) *calcd.* 312.1546, *found* 312.1557.
3f: HRMS (ESI) [C₁₆H₂₅NSi] (M+H)⁺ *calcd.* 258.1678, *found* 258.1680
3k: HRMS (EI) [C₁₅H₂₃NSi] (M⁺) *calcd.* 245.1600, *found* 245.1596.
3n: HRMS (EI) [C₁₃H₂₂Si] (M⁺) *calcd.* 206.1491, *found* 206.1487.
3o: HRMS (EI) [C₁₃H₂₂OSi] (M-CH₂CH₃)⁺ *calcd.* 193.1049, *found* 193.1044.
3cc: HRMS (EI) [C₁₀H₁₂F₃Si] (M-CH₂CH₃)⁺ *calcd.* 217.0660, *found* 217.0664.
5d: HRMS (EI) [C₁₇H₃₀Si] (M⁺) *calcd.* 262.2117, *found* 262.2111.
5e: HRMS (EI) [C₁₄H₂₁F₃OSi] (M⁺) *calcd.* 290.1314, *found* 290.1320
5f: HRMS (EI) [C₁₄H₂₄Si] (M⁺) *calcd.* 220.1647, *found* 220.1654.
5g: HRMS (EI) [C₁₇H₂₇Si] (M-CH₂CH₃)⁺ *calcd.* 259.1882, *found* 259.1884.
5h: HRMS (ESI) [C₁₆H₂₈NaSi] (M+Na)⁺ *calcd.* 271.1858, *found* 271.1857.
5i: HRMS (EI) [C₁₉H₂₄Si] (M⁺) *calcd.* 280.1647, *found* 280.1653.
5j: HRMS (ESI) [C₁₃H₂₃NSi] (M+H)⁺ *calcd.* 222.1678, *found* 222.1683
5l: HRMS (ESI) [C₁₆H₂₈NaSi] (M+Na)⁺ *calcd.* 271.1858, *found* 271.1857.
S7: HRMS (ESI) [C₃₂H₃₇FNO₄] (M+H)⁺ *calcd.* 518.2707, *found* 518.2683.
8: HRMS (EI) [C₃₂H₃₈FNO₄] (M⁺) *calcd.* 519.2785, *found* 519.2788.
9: HRMS (ESI) [C₃₈H₅₄NO₄Si] (M+H)⁺ *calcd.* 616.3822, *found* 616.3826.
16: HRMS (EI) [C₁₆H₂₆Si] (M⁺) *calcd.* 246.1804, *found* 246.1809.

Reviewer #2 (Remarks to the Author):

I have read the revised manuscript submitted the authors and confirmed that they considered the reviewers' opinions and clearly answered to all. I believe the revision improved the quality of this manuscript, now becoming to deserve publication in the Nature Communication. Before acceptance, I would ask the authors for tiny revisions listed below.

-- Fig. 3.: The circle of "°C" in the equation (c) should be superscript.

-- Fig. 4.: The equation (2) and the results shown in the small table is difficult to understand because the length of the alkyl chain in the starting materials is different; the "n" is 4 only for the alkyl fluoride to afford 5I, while the "n" for other halides is 3. Addition of an indication of the number "n" below the starting halides, such as "X = F: n = 3, X = Cl, Br, I: n = 3", would help the readership to understand the results. Addition of a footnote of the table to explain how to determine the ratio of the borylated product to the silylated product is also preferable.

-- Fig. 4.: The presentation of the ratio of the solvent in both equations should not be written in subscript.

Point-by-point responses to the reviewers' requests for "Defluorosilylation of Fluoroarenes and Fluoroalkanes" (NCOMMS-18-20843A)

Reviewer #2: (Remarks to the Author)

I have read the revised manuscript submitted the authors and confirmed that they considered the reviewers' opinions and clearly answered to all. I believe the revision improved the quality of this manuscript, now becoming to deserve publication in the Nature Communication. Before acceptance, I would ask the authors for tiny revisions listed below.

Response: Thank you!

-- Fig. 3.: The circle of "°C" in the equation (c) should be superscript.

Response: Revised.

-- Fig. 4.: The equation (2) and the results shown in the small table is difficult to understand because the length of the alkyl chain in the starting materials is different; the "n" is 4 only for the alkyl fluoride to afford 5l, while the "n" for other halides is 3. Addition of an indication of the number "n" below the starting halides, such as "X = F: n = 3, X = Cl, Br, I: n = 3", would help the readership to understand the results. Addition of a footnote of the table to explain how to determine the ratio of the borylated product to the silylated product is also preferable.

-- Fig. 4.: The presentation of the ratio of the solvent in both equations should not be written in subscript.

Response: All were revised.